# Complex of Lauric Acid Monoglyceride and Cinnamaldehyde Ameliorated Subclinical Necrotic Enteritis in Yellow-Feathered Broilers by Regulating Gut Morphology, Barrier, Inflammation and Serum Biochemistry

**DOI:** 10.3390/ani13030516

**Published:** 2023-02-01

**Authors:** Chaojun Zheng, Gengsheng Xiao, Xia Yan, Ting Qiu, Shun Liu, Jiancun Ou, Mingzhu Cen, Li Gong, Jiansheng Shi, Huihua Zhang

**Affiliations:** 1School of Life Science and Engineering, Foshan University, Foshan 528000, China; 2Laboratory of Livestock and Poultry Breeding, Institute of Animal Science, Guangdong Academy of Agricultural Sciences, Guangzhou 510000, China; 3College of Life Science and Food Engineering, Hebei University of Engineering, Handan 056038, China

**Keywords:** cinnamaldehyde, lauric acid monoglyceride, necrotic enteritis, gut morphology, gut barrier, plant essential oil

## Abstract

**Simple Summary:**

Under the background of total prohibition, it has always been the goal of the broiler industry to improve the harm of necrotizing enteritis and the performance of broiler chickens. As a potential green and healthy feed additive, lauric acid monoglyceride and cinnamaldehyde have important research significance. The results showed that 350 and 500 mg/kg combined plant essential oils of lauric acid monoglyceride and cinnamaldehyde could increase ADG and F:G, improve intestinal morphology and antioxidant capacity, and downregulate the expression of inflammatory factors.

**Abstract:**

This experiment investigated the benefits of plant essential oil (EO) composed with lauric acid monoglyceride and cinnamaldehyde on necrotic enteritis-challenged broilers. A total of 180 1-day-old healthy yellow-feathered broilers were randomly divided into 4 groups with 3 replicates of 15 chicks each. The experimental groups were as follows: the control group (CON) was fed with the basal diet and was not challenged by *Eimeria acervulina* (EA) and *Clostridium perfringens* (CP); CPEA group was also fed with a basal diet, but infected with CP and EA; CPEA_EO350 group and CPEA_EO500 group were fed with a basal diet supplemented with 350 and 500 mg/kg EO, respectively, and all infected with CP and EA. On the 7th day, each bird in the CPEA group, CPEA_EO350 group and CPEA_EO500 group was orally administrated with 1 mL *Eimeria acervulina* containing 5000 oocytes/mL, and the birds of the CON group were orally administrated with 1 mL normal saline. From the 15th day, 1 mL of CP type A CVCC-2030 strain (about 5 × 10^8^ cfu/mL) was orally inoculated into each bird of the CPEA, CPEA_EO350 and CPEA_EO500 groups for three consecutive days. Similarly, the CON group was orally given 1 mL of normal saline. The CPEA stimulation reduced the average daily gain (ADG) of broilers, increased the feed-to-gain ratio (F:G), and increased the intestinal lesions of the broilers (*p* < 0.01), indicating that CPEA stimulation was clinically successful. Compared with the CPEA group, the ADG of CPEA_EO350 and CPEA_EO500 increased, the F:G decreased (*p* < 0.01), and the intestinal score of CPEA_EO500 decreased (*p* < 0.01). The expression of the tight junction protein of the jejunum and ileum on 21d was upregulated (*p* < 0.01), and the expression of jejunum inflammation factors TNF-α on 21d and jejunum and ileum inflammatory factor IL-6 on 28d were also downregulated. The CPEA_EO350 and CPEA_EO500 increased antioxidant capacity. To sum up, 350 and 500 mg/kg of lauric acid monoglyceride and cinnamaldehyde complex plant essential oils can improve ADG and F:G, improve intestinal morphology and the body’s antioxidant capacity, and downregulate the expression of inflammatory factors. The concentration of 500 mg/kg performed even better.

## 1. Introduction

Chicken necrotic enteritis (NE) is a common intestinal bacterial disease in poultry, which brings huge economic losses to the global poultry industry every year [1]. NE is an intestinal bacterial disease present in fast-growing broiler flocks, mainly caused by NetB-producing strains of *Clostridium perfringens*, combined with one or more factors such as *Eimeria* [2], high-protein fish meal, etc., to cause subclinical or clinical disease [3]. Feeding antibiotics have a significant effect on the treatment of broilers and improving the growth of livestock and poultry, and they improve the efficiency of animal production, but at the same time, the problem of bacterial resistance caused by this has become more and more serious, and drug residues have caused certain harm to the environment and human health. China also banned the addition of feed antibiotics to feed in 2020 after the Eureopean Union and the United States banned them. It is leading to a dramatic increase in the global incidence of NE [1]. The ban on antibiotics makes it necessary for people to find suitable alternatives to antibiotics.

Many natural compounds found in plants have been shown to have antibacterial properties, antioxidant and anti-inflammatory capacities and to increase appetite and activity of endogenous enzymes [4]. They are receiving increasing attention as potential antibiotic replacements. However, their different chemical compositions and concentrations exert different antibacterial functions [5]. Therefore, in recent years, more and more people have developed plant essential oil as a potential alternative for feed antibiotics [6]. Lauric acid monoglyceride, a natural compound with antifungal, antibacterial, and antiviral activity, is also a bactericidal against a variety of potential bacterial pathogens [7]. Lauric acid monoglyceride has strong in vitro antibacterial activity, especially against Gram-positive microorganisms and *Clostridium perfringens* [8]. In vivo studies have also found the lowest incidence and severity of NE with the addition of a combination of lauric and butyric acid compared to other treatments [7]. Cinnamaldehyde is an aldehyde organic compound, which is abundant in cinnamon and other plants, and has strong antioxidant, analgesic, anti-ulcer and anti-heart damage activities. Several studies have demonstrated the effect of cinnamaldehyde on gut health and growth performance in broilers, but their findings have been inconsistent [9]. The type and concentration of the active compound may be responsible for the different results in the literature [10]. However, it has a good bacteriostatic effect on *Clostridium perfringens*, *Pseudomonas aeruginosa*, fecal Escherichia coli, *Staphylococcus aureus*, and *Escherichia coli* [11].

Lauric acid monoglyceride and cinnamaldehyde have achieved good results in the production performance and bacteriostatic effect of broilers. Studies have also shown that the synergistic effect of the combination of two or more compound components is greater than the sum of the effects of each single compound at the same dose [12]. In addition, specific complexes of plant essential oils appear to control *Eimeria acervulina* and *Clostridium perfringens* infections, potentially reducing the occurrence of necrotizing enteritis [10]. Although monoglyceride laurate and cinnamaldehyde have been used individually in broiler production as well as to eliminate pathogens, little is known about their combined effects [13]. Therefore, the objective of this study was to evaluate the effects of complexes (lauric acid monoglyceride and cinnamaldehyde) on growth performance, intestinal morphology, and immune barrier factors in yellow-feathered broilers infected with *Eimeria acervulina* and *Clostridium perfringens*.

## 2. Materials and Methods

### 2.1. Animals, Group Formation and Diets

A total of 180 one-day-old healthy male yellow-feathered broilers (Muyuan Animal Husbandry Co., Ltd., Guangzhou, China) were randomly divided into 4 four groups with 3 replicates of 15 chicks each. The groups were as follows: control group (CON) was fed with a basal diet (corn-soybean meal type, the feed ingredients and dietary nutrient compositions are presented Table 1) and not challenged with *Eimeria acervulina* (EA) and *Clostridium perfringens* (CP); CPEA group was also fed with a basal diet, but infected with EA and CP; CPEA_EO350 group and CPEA_EO500 group were fed with a basal diet supplemented with 350 and 500 mg/kg plant essential oils (EO) (Jiabiyou Biotechnology Co., Ltd., Wuhan, China), respectively, and were infected with EA and CP. The compositions of plant EO are 15% cinnamaldehyde and 85% monoglyceryl laurate. Fifteen chicks per replicate were assigned to one cage (65 × 60 × 40 cm) throughout the experimental period. The chicks were housed in environmentally controlled rooms with fresh water and feed provided ad libitum. The room temperature was maintained at 33 to 34 °C for the first three days, then decreased by 3 °C each week to reach a final temperature of 24 °C. The first week of light was 23 h, after which it was reduced to 16 h.

### 2.2. Eimeria acervulina and Clostridium perfringens Infection Protocol

At day 7, each bird in the CPEA group, CPEA_EO350 group and CPEA_EO500 group was orally administrated 1 mL *Eimeria acervulina*, containing 5000 oocytes/mL (Zhengdian Biotechnology Co., Ltd., Foshan, China); birds in the CON group were orally administrated 1 mL/bird normal saline. Then, starting at day 15, 1 mL of *Clostridium perfringens* type ACVCC-2030 strain (about 5 × 10^8^ cfu/mL) (China Culture Collection of Microorganisms, Beijing, China) was orally inoculated to each bird of the CPEA group, CPEA_EO350 group and CPEA_EO500 group for three consecutive days. Similarly, the birds in the CON group were given 1 mL/bird of normal saline. In the event of any death following *Eimeria acervulina* and *Clostridium perfringens* challenge, an autopsy was performed to determine the cause of death. Repeated body weight (BW), average daily feed intake (ADFI) and average daily gain (ADG) were measured on days 21 and 28, and the feed-to-gain ratio was calculated accordingly, taking into account mortality.

### 2.3. Intestinal Pathological Damage

At 28d, 3 birds were randomly selected for each replicate, and a total of 9 birds in each group were selected for intestinal pathological damage scoring. Intestinal pathological damage scores were determined by two experienced researchers using the 0 to 5 damage scale [14]. In short, 0 point: normal without necrotizing enteritis; 1 point: the intestinal wall is flabby and thin, and the intestinal body cannot be restored to its original state after the fingers are pressed; 2 point: mild necrotizing enteritis with few (1–6) punctate ulcers and gangrene; 3 point: moderate necrotizing enteritis with more (more than 6) punctate or small areas of ulceration and gangrene; 4 point: severe necrotizing enteritis, extensive ulceration and gangrene, hemorrhage, partial necrosis of the intestine; 5 point: dead or dying.

### 2.4. Intestinal Tissue Section

For each repetition, two birds were selected randomly, resulting in six birds per group. The jejunum and ileum tissue specimens of euthanized chickens (euthanasia by artificial neck dislocation) were immersed in 4% formaldehyde solution for fixation. A paraffin embedding process followed after dehydrating and clearing intestinal tissue. The paraffin-embedded sections were cut into five serial sections, rehydrated and stained with hematoxylin and eosin (H&E) after four serial sections were rehydrated. Tissue sections were observed using a LEICA (DFC290 HD system digital camera, Heilbrugger, Switzerland) connected to an optical microscope with 10 objective lenses. The villus height and the corresponding crypt depth of straight and relatively intact villos were measured.

### 2.5. RNA Extraction and cDNA Synthesis

On 21d and 28d, the jejunal and ileal mucosa samples of euthanized chickens were scraped and placed in RNAase-free tubes, immediately placed in liquid nitrogen, and stored at −80 °C. The total RNA of each sample was extracted with TransZol reagent (TransGen Biotech, Beijing, China), and the concentration of RNA was measured with a VWRI732-2534 spectrophotometer (Avantor, Radnor, PA, USA), and 1 μL of the extracted total RNA solution was taken to read the RNA concentration, OD260/280 and 260/230 ratio, the ratio is in the range of 1.8~2.0, it is considered that the RNA sample has high integrity and meets the requirements of the next experiment. According to the method of RT EasyTMⅡ (With gDNase) kit (Foregene, Chengdu, China), 1 μg of RNA was taken to remove genomic DNA contamination and reverse transcribed to synthesize cDNA, which was stored at −20 °C until required for quantitative PCR.

### 2.6. Quantitative PCR

The primer pairs were designed for specific genes by the NCBI primer tool (http://www.ncbi.nlm.nih.gov, accessed on 1 December 2022), and the primer pairs were specifically detected. The primer sequences of the genes are shown in Table 2. The primers were synthesized by Qingke Biological Co., Ltd. cDNA gene expression assays were performed on real-time quantitative PCR (QuantStudio 3, Thermo Fisher Scientific, Shanghai, China) using Real-Time PCR EasyTM-SYBR Green I kit (Foregene, Chengdu, China). The detected immune-related gene screens and barrier structure-related genes are Mucin-2 (MUC-2), ZO-2, ZO-1, claudin-3, occludin, interleukin-6 (IL-6), interleukin-17A (IL-17A), interleukin-22 (IL-22), tumor necrosis factor-α (TFN-α), interferon-β (IFN-β), interferon-γ (IFN-γ) with β-actin as the internal reference gene and the control group as the reference; the relative expression of the gene was calculated by the 2^−ΔΔCt^ method.

### 2.7. Serum Biochemical Indicators

At 21d and 28d, each replicate had two birds randomly selected. Pterygoid vein blood samples were collected after fasted 12 h and the serum was centrifuged at 3000× *g* for 15 min at 4 °C after 2 h of quiescence. Using Nanjing Jiancheng Bioengineering Institute kits (Nanjing, Jiangsu, China), serum concentrations were determined: total protein (TP), albumin (ALB), uric acid (UA), urea nitrogen (BUN), alanine aminotransferase (ALT), aspartate aminotransferase (ASL), superoxide dismutase (SOD), total antioxidant capacity (T-AOC), and malondialdehyde (MDA). 

### 2.8. Statistical Analysis 

In SPSS 25.0 (SPSS, Inc., Chicago, IL, USA), independent samples t test was used to analyze all data, and all data were expressed as means ± SD. The model includes fixed effects, and GraphPad Prism version 9 (GraphPad Software, La Jolla, CA, USA) was used to present it. A *p* value < 0.05 was considered significant (*p* < 0.05, *p* < 0.01), and 0.05 < *p* value < 0.10 was discussed as tendencies.

## 3. Results

### 3.1. Growth Performance

Compared with the CON group, the *Eimeria acervulina* and *Clostridium perfringens* challenge reduced the BW of CPEA at 21d and 28d (*p* < 0.01; Table 3) and improved the feed-to-gain ratio during 1–28d (*p* < 0.01; Table 3). Compared with the CPEA group, the CPEA_EO500 and CPEA_EO350 groups increased the BW at 28 by 7% and 5.8%, respectively (*p* < 0.01; Table 3). Compared with the CPEA group, the CPEA_EO500 and CPEA_EO350 groups decreased the ratio of feed to gain during 1–28d by 0.25 and 0.21, respectively (*p* < 0.01; Table 3).

### 3.2. Intestinal Pathology Score

At 21d, there was no difference in intestinal scores among the four groups (*p* > 0.05; Figure 1); At 28d, the lesion score of the CPEA group was significantly higher than those of the CON group (*p* < 0.01; Figure 1) and CPEA_EO500 group (*p* < 0.05; Figure 1). The intestinal score of the CPEA_EO350 group showed a decreasing trend compared with the CPEA group.

### 3.3. Intestinal Tissue Morphology

Combining the slice pictures and the analysis of the histogram, compared with the CON group, it can be seen that the jejunum villus height of the CPEA group was decreased on 21d and 28d (*p* < 0.01; Figure 2). The height of the jejunal villi in the CPEA_EO500 group was increased (*p* < 0.01; Figure 2), and there was a trend of increasing jejunal villus height in the CPEA_EO350 (*p* < 0.10; Figure 2).

### 3.4. Intestinal Mucosal Barrier Function

Compared with the CON group, the CPEA group downregulated the mRNA expressions of MUC-2, ZO-2, ZO-1, claudin-3, occludin in the jejunum and ileum at 21d (*p* < 0.01; Table 4). Compared with the CPEA group, the CPEA_EO350 and CPEA_EO500 groups upregulated the mRNA expressions of MUC-2, ZO-2, ZO-1, claudin-3 and occludin in the jejunum and ileum (*p* < 0.10; Table 4). On the 28th day, the CPEA group downregulated the mRNA expression of jejunal MUC-2, ZO-1, claudin-3 and ileal MUC-2, ZO-2 (*p* < 0.01; Table 4). Compared with the CPEA group, the CPEA_EO350 and CPEA_EO500 groups upregulated the mRNA expressions of ZO-1 in the jejunum and ZO-2 and claudin-3 in the ileum (*p* < 0.01; Table 4), and the CPEA_EO350 group also upregulated the mRNA expressions of ZO-1 in the jejunum and ZO-2 and claudin-3 in the ileum (*p* < 0.01; Table 4). In addition, CPE-CML350 upregulated the mRNA expression of Claudin-3 in the jejunum and the mRNA expression of ZO-1, Occludin and MUC-2 in the ileum at 28d compared with the CPEA (*p* < 0.01; Table 4).

### 3.5. Intestinal Mucosal Inflammatory Factors

Compared with the CON group, the mRNA expressions of IL-6, IL-17 and TNF-α in the jejunum and IL-6 and IL-17 in the ileum were upregulated in the CPEA group at 21d (*p* < 0.01; Table 5), while the CPEA_EO350 and CPEA_EO500 groups had downregulated TNF-α and IL-6 and IL-17 mRNA expressions in the jejunum compared with the CPEA group (*p* < 0.01; Table 5). The mRNA expression of IL-17 in the jejunum of the CPEA_EO350 group was downregulated by 0.99 (*p* < 0.01; Table 5). Compared with the CON group, the mRNA expression of IL-6 in the jejunum and ileum of the CPEA group was upregulated (*p* < 0.01; Table 5). While the mRNA expression of IL-6 in the jejunum and ileum of the CPEA_EO350 and CPEA_EO500 groups was downregulated (*p* < 0.01; Table 5), and the mRNA expression of IFN-γ in the CPEA_EO350 group was downregulated by 0.76 compared with the CEPA group (*p* < 0.01; Table 5).

### 3.6. Serum Biochemical Indices

Compared with the CPEA group, the serum TP content in the CPE-CML500 group was significantly increased on 21d (*p* < 0.01; Table 6). The serum BUN content in the CPE-CML350 group and CPE-CML500 group was significantly increased compared with the CON group on 21d (*p* < 0.01; Table 6). Compared with the CPEA group, MDA content in the serum of the CPE-CML350 group and CPE-CML500 group was significantly decreased and SOD content was significantly increased on 21d (*p* < 0.01; Table 6). Compared with the CPEA group, T-AOC content in the serum of the CPE-CML500 group was significantly increased on 21d (*p* < 0.01; Table 6).

Compared with the CON group, the serum TP content in the CPEA group was significantly decreased on 28d (*p* < 0.01; Table 6), but compared with the CPEA group, serum TP content in the CPE-CML500 group and CPE-CML350 group had an increasing trend (*p* < 0.05; Table 6). Compared with the CPEA group, MDA content in the serum of the CPE-CML500 group and CPE-CML350 group was significantly decreased, while SOD content was significantly increased (*p* < 0.01; Table 6)

## 4. Discussion

This study showed that broiler chickens successfully developed NE after challenged with *Eimeria acervulina* and *Clostridium perfringens*, which was characterized by deterioration of growth performance and intestinal necrosis. It has been reported that broilers with NE have lower body weight gain, higher feed-to-gain ratio, and lower feed utilization efficiency [15]. Therefore, in recent years, people have been exploring the use of feed additives to combat NE to reduce its burden and economic losses. In the current experiment, concentrations of 350 and 500 mg/kg lauric acid monoglyceride and cinnamaldehyde complex increased the average daily gain of challenge broilers and decreased the feed-to-gain ratio. Al-Kassie [16] showed that the weight gain of broilers fed the dietary additive supplemented with cinnamaldehyde was significantly higher than that of the control group (without the addition of vegetable essential oils), and the feed-to-gain ratio was also lower than that of the control group (without the addition of vegetable essential oils). Shirzadegan [17] observed that the addition of different concentrations of cinnamon powder to the diet significantly increased the final body weight of broilers. In addition, it was found that adding cinnamaldehyde and celery seed oil to the broiler diet at the same time increased the 42-day-old body weight of broilers [18]. However, Koochaksaraie, Irani [19] showed that adding cinnamon powder has no significant effect on the growth of broilers. Adding cinnamaldehyde to the feed had no significant improvement in the weight gain of broilers, but it significantly reduced water intake [20]. This is likely to be the result of different concentrations of essential oils or types of complexes.

It has been reported that plant essential oils can improve the production performance of broilers and effectively control the proliferation of *Clostridium perfringens* [21,22]. In addition, the anti-coccidial effect of essential oils has also been reported [23]. It appears that the cinnamic aldehyde and lauric acid complexes at concentrations of 350 and 500 mg/kg were also efficient to reduce *Eimeria acervulina* and *Clostridium perfringens* attacks.

The villi and crypts are important parts of the small intestine, and their height and depth can reflect the absorptive capacity of the small intestine [8]. Our results showed that the addition of lauric acid monoglyceride and cinnamaldehyde complex to the feed improves the villus height in the jejunum and ileum, which is consistent with previous findings [24,25]. Studies have also shown that some enzymes or enzymes in combination with dietary components can improve the absorption surface of the small intestine, affecting the intestinal barrier of chickens [26]. The mechanical barrier composed of intestinal mucosal epithelial cells and their tight junction proteins is particularly important in the intestinal barrier [27]. Claudin has a variety of functions, not only maintaining the epithelial barrier function, preventing the invasion of toxic macromolecules and microorganisms, but also selectively regulating the entry of small molecules and ions into the body [28]. A tight junction is a highly dynamic structure, and its degree of closure changes depending on external stimuli and physiological and pathological factors [29]. Claudin, occludin and ZO are an important family of intestinal tight junction cells [30]. Our study found that in the state of NE, the expression of barrier factors MUC-2, ZO-2, ZO-1, claudin-3, and occludin in the tight junction structure of the jejunum and ileum were downregulated at 21d, while after NE stimulation, the expression of MUC-2, ZO-2, ZO-1, claudin-3, and occludin could be upregulated when lauric acid monoglyceride and cinnamaldehyde complex were supplemented, indicating that the complex of lauric acid monoglyceride and cinnamaldehyde has a positive effect on tight junction cells or intestinal physical barriers. Therefore, the improvement of the small intestine morphology may also be one of the reasons for the increase in the digestibility of nutrients and the decrease in the ratio of feed to gain.

MDA is the end product of lipid peroxidation produced in reaction with free radicals and is a useful biomarker of oxidative stress in vivo [31]. The content of MDA in serum challenged by *Eimeria acervulina* and *Clostridium perfringens* was decreased by both the CPEA_EO350 group and CPEA_EO500 group, and the content of superoxide dismutase and the total antioxidant capacity of serum in the CPEA_EO500 group also increased. This is consistent with the results of plant-derived substances, which can improve the antioxidant activity of broilers and reduce the content of MDA in serum [32]. Plant essential oils such as eucalyptus also have free radical scavenging and anti-oxidative effect abilities [33]. Lauric acid monoglyceride and cinnamaldehyde are natural compounds that have free radical scavenging activity to protect lipids from oxidation, thereby delaying the process of lipid peroxidation (Belasli et al., 2020). The extracts have high phenolic content and excellent free radical scavenging potential, which can remove excess free radicals because their phenolic hydroxyl groups act as hydrogen donors for surrogate free radicals generated during the initial lipid oxidation process [34,35]. Therefore, it reduces the formation of hydroxyl peroxide, thereby reducing the content of malondialdehyde in the body and improving the total antioxidant capacity of the body [36]. There are also some studies suggesting that the mechanism by which essential oils enhance antioxidant capacity may be attributed to their ability to donate hydrogen or electrons and to delocalize unpaired electrons within the aromatic structure [37].

Bioactive substances with antioxidant potential have been proven to have anti-inflammatory properties [38], and it can also be seen from the experimental results that the addition of lauric acid monoglyceride and cinnamaldehyde complexes to the feed downregulates the pro-inflammatory factor IL- 6 in the jejunum and ileum at day 28, and the lauric acid monoglyceride and cinnamaldehyde complexes at a concentration of 500 mg/kg had a greater tendency to downregulate. The pro-inflammatory factor TNF-α was also downregulated in the jejunum at day 21, suggesting that the lauric acid monoglyceride and cinnamaldehyde complex can alleviate the inflammatory response caused by NE. Krauze, Cendrowska-Pinkosz [39] showed that with continuous or regular addition of phytobiotic supplements containing cinnamon oil, broiler body IL-6 levels are beneficially reduced, and the concentration of lysozymes is also increased. 

The total protein content in serum was increased by the addition of 500 mg/kg of lauric acid monoglyceride and cinnamaldehyde complex in the present study. Amadbr and Zentek [40] reported that a plant-derived mixed essential oil feed additive significantly improved total protein content in broiler serum. Ghazalah and Ali [41] also reported that the serum protein and albumin levels of broilers were increased after the addition of plant rosemary leaf essential oil to broiler feed. While feed supplemented with lauric acid monoglyceride and cinnamaldehyde complex increased total protein in serum, which may be related to weight gain in broilers [41], the increase in total protein content in animal serum may also indicate enhanced nutrient supply and transport. The addition of lauric acid monoglyceride and cinnamaldehyde complex in the feed also increased the content of urea nitrogen in serum, which may also be related to the increase in total protein in serum. Serum urea nitrogen levels decreased, possibly due to increased protein catabolism, as albumin and total protein levels also decreased, while feeding patterns were unchanged [42].

However, the disadvantage of this study is that only two concentration gradients of 350 and 500 mg/kg were created, and it is impossible to judge whether the higher concentration dose will show better effects or show harmful effects. Whether the different proportions of lauric acid monoglyceride and cinnamaldehyde will cause different results needs further study.

## 5. Conclusions

In this study, supplementation of 350 and 500 mg/kg of cinnamaldehyde and lauric acid complexs in diets can improve the harm caused by necrotizing enteritis, and 500 mg/kg is better in improving performance, intestinal pathological damage score and intestinal morphology of yellow-feathered broilers. The disadvantage of this study is that only two concentration gradients of 350 and 500 mg/kg were studied, and it is not possible to judge whether higher concentration doses will show better results or show harmful effects retained cinnamaldehyde and lauric acid. Do different ratios of cinnamaldehyde and lauric acid lead to different results? These are all worthy of further study.

## Figures and Tables

**Figure 1 animals-13-00516-f001:**
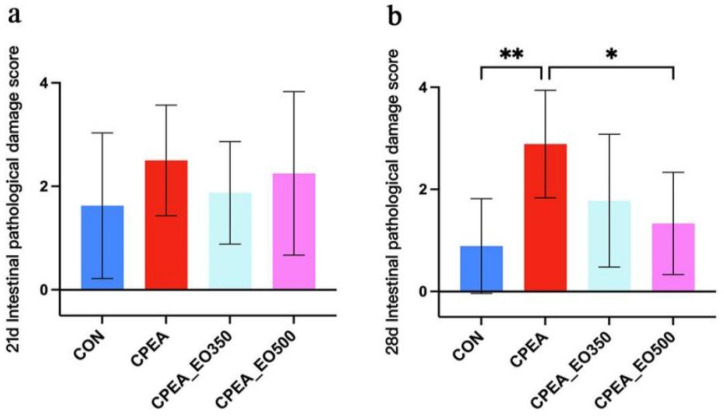
(**a**) 21d Intestinal pathological damage score; (**b**) 28d Intestinal pathological damage score. CON, control group birds fed a basal diet; CPEA, birds fed a basal diet and infected with Clostridium perfringens and Eimeria acervuline; CPEA-EO350, birds fed a basal diet supplemented with 350 mg/kg essential oils and infected with Clostridium perfringens and Eimeria acervuline; CPEA-EO500, birds fed a basal diet supplemented with 500 mg/kg essential oils and infected with Clostridium perfringens and Eimeria acervuline. Significant deviations are denoted by asterisks (* *p* < 0.05, ** *p* < 0.01).

**Figure 2 animals-13-00516-f002:**
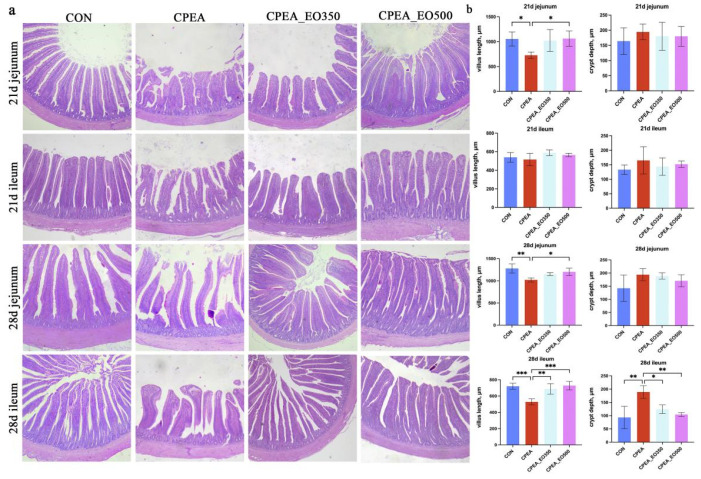
(**a**) Jejunal and ileal mucosa photomicrograph with typical crypts and villi. The scale bar is 200 μm. Eosin and haematoxylin stain. Magnification is 100×. (**b**) Crypt depth and villus height of jejunum and ileum. CON, control group birds fed a basal diet; CPEA, birds fed a basal diet and infected with Clostridium perfringens and Eimeria acervuline; CPEA-EO350, birds fed a basal diet supplemented with 350 mg/kg essential oils and infected with Clostridium perfringens and Eimeria acervuline; CPEA-EO500, birds fed a basal diet supplemented with 500 mg/kg essential oils and infected with Clostridium perfringens and Eimeria acervuline. Significant deviations are denoted by asterisks (* *p* < 0.05, ** *p* < 0.01, *** *p* < 0.001).

**Table 1 animals-13-00516-t001:** Basic diet ingredients and nutritional composition.

Feed Ingredients	%	Nutrient Composition	%
Corn	61.91	ME (kcal/kg)	2934.95
Bean pulp	26.00	Crude protein	20.31
Soybean meal	3.00	Calcium	0.87
Corn gluten meal	3.00	Phosphorus	0.66
Soybean oil	1.50	Available phosphate	0.41
Limestone	2.00	Lysine	1.01
		DL-Methionine	0.44
Dicalcium	1.65	Methionine and Cysteine	0.75
Salt	0.35	Threonine	0.73
Mineral premix	0.2		
Phytase	0.02		
Vitamin premix	0.03		
Choline chloride	0.12		
Lysine-HCL	0.08		
DL-Methionine	0.14		
Total	100		

Mineral premix includes the following: Fe, 80 mg; Cu, 8 mg; Mn, 100 mg; Zn, 80 mg; I, 10.7 mg; Se, 0.3 mg; vitamin A, 6000 IU; vitamin D, 1000 IU; vitamin E, 10 IU; vitamin K, 0.5 mg/kg; thiamine, 2 mg/kg; riboflavin, 5 mg/kg; pantothenic acid, 10 mg/kg; niacin, 30 mg/kg; pyridoxine, 3 mg/kg; biotin, 0.15 mg/kg; folic acid, 0.55 mg/kg; vitamin B, 120.01 mg/kg.

**Table 2 animals-13-00516-t002:** The primer sequence of the gene.

Gene	Sequence (5′–3′)	Ta °C
MUC-2	F: TTCATGATGCCTGCTCTTGTG	57.92
	R: CCTGAGCCTTGGTACATTCTTGT	
ZO-2	F: AGTGGCCACCATTGTTGTGA	55.58
	R: ACTGTAGCCACTTCGAGCAC	
ZO-1	F: CTTCAGGTGTTTCTCTTCCTCCTC	56.61
	R: CTGTGGTTTCATGGCTGGATC	
Claudin-3	F: CGGGATTTCTACAACCCGCT	57.65
	R: GAGTAAGCCACCTTGCTGGG	
Occludin	F: CGGAGCCCAGACTACCAAAG	55.84
	R: TTACACAGCTTCAGCCTTACA	
IL-6	F: CAGGACGAGATGTGCAAGAA	56.75
	R: TAGCACAGAGACTCGACGTT	
IL-17A	F: GAAGGTGATACGGCCAGGAC	56.78
	R: TGGGTTAGGCATCCAGCATC	
IL-22	F: GCCCTACATCAGGAATCGCA	57.87
	R: TCTGAGAGCCTGGCCATTTC	
TNF-α	F: TGTGTATGTGCAGCAACCCGTAGT	57.94
	R: GGCATTGCAATTTGGACAGAAGT	
IFN-β	F: TGCAACCATCTTCGTCACCA	56.68
	R: GGAGGTGGAGCCGTATTCTG	
IFN-γ	F: ACACTGACAAGTCAAAGCCGC	58.66
	R: AGTCGTTCATCGGGAGCTTG	

**Table 3 animals-13-00516-t003:** The growth performance of broilers.

Item	CON	CPE A	CPE-CML350	CPE-CML500	SEM	*p* Value
BW, g						
1d	39.20 ± 0.15	39.39 ± 0.13	39.22 ± 0.11	39.19 ± 0.19	0.04	0.35
21d	378.57 ± 7.14 ^a^	242.86 ± 7.14 ^c^	269.05 ± 23.24 ^b^	277.86 ± 3.71 ^b^	15.86	<0.01
28d	635.19 ± 6.41 ^a^	484.82 ± 4.49 ^c^	514.82 ± 8.48 ^b^	521.48 ± 5.7 ^b^	17.31	<0.01
ADFI, g/d						
1 to 21d	25.79 ± 0.56 ^a^	23.05 ± 0.83 ^c^	24.29 ± 0.47 ^b^	23.32 ± 0.51 ^bc^	0.36	<0.01
21 to 28d	62.11 ± 0.47 ^b^	68.77 ± 1.5 ^a^	60.42 ± 2.63 ^b^	62.45 ± 2.11 ^b^	1.06	<0.01
1 to 28d	35.02 ± 0.71 ^a^	34.9 ± 0.99 ^ab^	33.66 ± 0.58 ^bc^	33.47 ± 0.33 ^c^	0.27	0.04
ADG, g/d						
1 to 21d	16.97 ± 0.35 ^a^	10.17 ± 0.35 ^c^	11.49 ± 1.16 ^b^	11.95 ± 0.18 ^b^	0.79	<0.01
21 to 28d	36.66 ± 1.37	34.57 ± 1.66	35.11 ± 3.84	34.8 ± 0.34	0.60	0.66
1 to 28d	22.07 ± 0.24 ^a^	16.49 ± 0.17 ^c^	17.61 ± 0.32 ^b^	17.87 ± 0.21 ^b^	0.64	<0.01
F:G, g/g						
1 to 21d	1.53 ± 0 ^c^	2.27 ± 0.07 ^a^	2.13 ± 0.19 ^ab^	1.95 ± 0.02 ^b^	0.09	<0.01
21 to 28d	1.70 ± 0.05 ^b^	1.99 ± 0.11 ^a^	1.74 ± 0.22 ^b^	1.79 ± 0.06 ^ab^	0.05	0.09
1 to 28d	1.59 ± 0.02 ^c^	2.12 ± 0.07 ^a^	1.91 ± 0.07 ^b^	1.87 ± 0.02 ^b^	0.06	<0.01

Abbreviations: CON, control group birds fed a basal diet; CPEA, birds fed a basal diet and infected with Clostridium perfringens and Eimeria acervuline; CPEA-EO350, birds fed a basal diet supplemented with 350 mg/kg essential oils and infected with Clostridium perfringens and Eimeria acervuline; CPEA-EO500, birds fed a basal diet supplemented with 500 mg/kg essential oils and infected with Clostridium perfringens and Eimeria acervuline. BW, body weight; ADFI, average daily feed intake; ADG, average daily gain; F:G, feed-to-gain ratio. ^a–c^ Means within a column lacking a common superscript differ as per the corresponding *p* value indicated in the *p* value row.

**Table 4 animals-13-00516-t004:** Intestinal mucosal barrier function.

Item	CON	CPEA	CPE-CML350	CPE-CML500	SEM	*p* Value
21d jejunum						
MUC-2	1.00 ± 0 ^a^	0.46 ± 0.3 ^b^	0.65 ± 0.07 ^ab^	0.7 ± 0.41 ^ab^	0.08	0.07
ZO-2	1.00 ± 0 ^a^	0.52 ± 0.25 ^b^	0.68 ± 0.22 ^b^	0.56 ± 0.07 ^b^	0.06	<0.01
ZO-1	1.00 ± 0 ^a^	0.49 ± 0.42 ^b^	0.52 ± 0.31 ^b^	0.72 ± 0.08 ^ab^	0.08	0.06
Claudin-3	1.00 ± 0 ^a^	0.46 ± 0.25 ^c^	0.89 ± 0.22 ^ab^	0.63 ± 0.17 ^bc^	0.07	<0.01
Occludin	1.00 ± 0 ^a^	0.33 ± 0.28 ^b^	0.59 ± 0.23 ^b^	0.51 ± 0.22 ^b^	0.08	<0.01
21d ileum						
MUC-2	1.00 ± 0 ^ab^	0.41 ± 0.12 ^b^	1.12 ± 0.59 ^ab^	1.66 ± 1.2 ^a^	0.15	0.01
ZO-2	1.00 ± 0	1.25 ± 0.98	1.63 ± 1.01	1.49 ± 0.97	0.18	0.59
ZO-1	1.00 ± 0	0.98 ± 0.86	1.44 ± 0.47	1.45 ± 0.23	0.11	0.24
Cl audin-3	1.00 ± 0 ^b^	0.95 ± 0.4 ^b^	1.07 ± 0.27 ^b^	1.76 ± 0.46 ^a^	0.09	<0.01
Occludin	1.00 ± 0 ^b^	0.86 ± 0.3 ^b^	0.98 ± 0.27 ^b^	1.84 ± 0.31 ^a^	0.10	<0.01
28d jejunum						
MUC-2	1.00 ± 0 ^a^	0.52 ± 0.32 ^b^	0.71 ± 0.39 ^ab^	0.61 ± 0.27 ^b^	0.07	0.05
ZO-2	1.00 ± 0	0.91 ± 0.29	1.01 ± 0.51	0.65 ± 0.26	0.07	0.25
ZO-1	1.00 ± 0 ^a^	0.55 ± 0.25 ^b^	1.16 ± 0.51 ^a^	0.97 ± 0.35 ^a^	0.08	0.03
Claudin-3	1.00 ± 0 ^a^	0.57 ± 0.29 ^b^	0.99 ± 0.22 ^a^	0.58 ± 0.29 ^b^	0.06	<0.01
Occludin	1.00 ± 0	0.55 ± 0.58	0.64 ± 0.43	0.74 ± 0.37	0.08	0.27
28d ileum						
MUC-2	1.00 ± 0 ^a^	0.62 ± 0.23 ^b^	0.96 ± 0.37 ^a^	0.55 ± 0.31 ^b^	0.07	0.02
ZO-2	1.00 ± 0 ^a^	0.53 ± 0.05 ^b^	0.85 ± 0.07 ^a^	0.93 ± 0.34 ^a^	0.05	<0.01
ZO-1	1.00 ± 0 ^b^	0.98 ± 0.45 ^b^	1.69 ± 0.34 ^a^	1.41 ± 0.39 ^ab^	0.10	<0.01
Claudin-3	1.00 ± 0 ^ab^	0.45 ± 0.18 ^b^	1.14 ± 0.14 ^a^	1.43 ± 1.01 ^a^	0.12	0.03
Occludin	1.00 ± 0 ^b^	0.42 ± 0.28 ^b^	1.60 ± 0.84 ^a^	0.86 ± 0.06 ^b^	0.13	<0.01

Abbreviations: CON, control group birds fed a basal diet; CPEA, birds fed a basal diet and infected with Clostridium perfringens and Eimeria acervuline; CPEA-EO350, birds fed a basal diet supplemented with 350 mg/kg essential oils and infected with Clostridium perfringens and Eimeria acervuline; CPEA-EO500, birds fed a basal diet supplemented with 500 mg/kg essential oils and infected with Clostridium perfringens and Eimeria acervuline. ^a–c^ Means within a column lacking a common superscript differ as per the corresponding *p* value indicated in the *p* value row.

**Table 5 animals-13-00516-t005:** Intestinal mucosal inflammatory factors.

Item	CON	CPEA	CPE-CML350	CPE-CML500	SEM	*p* Value
21d jejunum						
IL-6	1.00 ± 0 ^b^	2.23 ± 0.46 ^a^	1.84 ± 0.7 ^ab^	1.66 ± 0.63 ^ab^	0.16	0.04
IL-17	1.00 ± 0 ^b^	2.25 ± 0.68 ^a^	1.26 ± 0.3 ^b^	1.81 ± 0.58 ^ab^	0.18	0.02
IL-22	1.00 ± 0 ^ab^	0.70 ± 0.19 ^b^	1.14 ± 0.49 ^a^	0.75 ± 0.11 ^ab^	0.08	0.10
TNF-α	1.00 ± 0 ^b^	2.10 ± 0.74 ^a^	1.19 ± 0.38 ^b^	1.22 ± 0.33 ^b^	0.15	0.02
IFN-β	1.00 ± 0	1.13 ± 0.58	1.24 ± 0.83	0.97 ± 0.41	0.13	0.89
IFN-γ	1.00 ± 0	1.62 ± 0.8	1.65 ± 0.8	1.49 ± 0.61	0.16	0.47
21d ileum						
IL-6	1.00 ± 0 ^b^	5.73 ± 1.02 ^a^	1.3 ± 1.42 ^b^	1.38 ± 0.69 ^b^	0.44	<0.01
IL-17	1.00 ± 0 ^c^	4.04 ± 1.26 ^a^	1.52 ± 0.19 ^bc^	2.06 ± 0.41 ^b^	0.29	<0.01
IL-22	1.00 ± 0 ^c^	1.69 ± 0.33 ^b^	2.82 ± 0.08 ^a^	2.58 ± 0.57 ^a^	0.19	<0.01
TNF-α	1.00 ± 0	1.71 ± 0.65	1.76 ± 0.7	1.47 ± 0.53	0.14	0.22
IFN-β	1.00 ± 0	2.03 ± 0.34	0.96 ± 1.27	2.09 ± 1.28	0.25	0.19
IFN-γ	1.00 ± 0 ^b^	2.54 ± 0.66 ^a^	1.45 ± 0.58 ^ab^	1.92 ± 1.1 ^ab^	0.22	0.05
28d jejunum						
IL-6	1.00 ± 0 ^c^	3.82 ± 0.96 ^a^	2.21 ± 0.6 ^b^	1.35 ± 0.87 ^bc^	0.32	<0.01
IL-17	1.00 ± 0 ^b^	2.34 ± 0.32 ^a^	2.04 ± 0.94 ^ab^	1.41 ± 1.22 ^ab^	0.22	0.10
IL-22	1.00 ± 0 ^b^	2.44 ± 0.67 ^a^	2.38 ± 0.73 ^a^	1.59 ± 1.46 ^ab^	0.25	0.09
TNF-α	1.00 ± 0 ^ab^	2.57 ± 0.36 ^a^	0.64 ± 0.35 ^b^	0.81 ± 0.34 ^b^	0.17	<0.01
IFN-β	1.00 ± 0 ^ab^	0.45 ± 0.38 ^b^	0.98 ± 0.35 ^ab^	1.66 ± 1.18 ^a^	0.15	0.04
IFN-γ	1.00 ± 0	1.22 ± 0.89	0.88 ± 0.54	0.89 ± 0.62	0.14	0.86
28d ileum						
IL-6	1.00 ± 0 ^b^	3.92 ± 1.51 ^a^	1.60 ± 0.87 ^b^	1.23 ± 0.87 ^b^	0.32	<0.01
IL-17	1.00 ± 0	1.72 ± 0.62	1.24 ± 0.65	1.02 ± 0.68	0.14	0.20
IL-22	1.00 ± 0	1.41 ± 0.76	1.75 ± 0.43	2.07 ± 1.25	0.17	0.22
TNF-α	1.00 ± 0 ^b^	1.94 ± 0.37 ^a^	1.30 ± 0.42 ^ab^	1.45 ± 0.85 ^ab^	0.13	0.04
IFN-β	1.00 ± 0 ^b^	0.87 ± 0.72 ^b^	1.19 ± 0.17 ^b^	2.51 ± 1.69 ^a^	0.24	0.04

Abbreviations: CON, control group birds fed a basal diet; CPEA, birds fed a basal diet and infected with Clostridium perfringens and Eimeria acervuline; CPEA-EO350, birds fed a basal diet supplemented with 350 mg/kg essential oils and infected with Clostridium perfringens and Eimeria acervuline; CPEA-EO500, birds fed a basal diet supplemented with 500 mg/kg essential oils and infected with Clostridium perfringens and Eimeria acervuline. ^a–c^ Means within a column lacking a common superscript differ as per the corresponding *p* value indicated in the *p* value row.

**Table 6 animals-13-00516-t006:** Serum biochemical indicators.

Item	CON	CPEA	CPE-CML350	CPE-CML500	SEM	*p* Value
21d						
TP, ug/ml	13.61 ± 3.99 ^ab^	11.32 ± 4.01 ^b^	15.58 ± 5.89 ^ab^	20.11 ± 3.33 ^a^	1.29	0.08
AIB, g/L	13.62 ± 1.72 ^a^	10.83 ± 0.5 ^b^	12.04 ± 1.39 ^ab^	10.18 ± 0.99 ^b^	0.44	0.01
UA, μmol/ml	379.06 ± 69.25	290.13 ± 121.42	258.85 ± 102.81	282.31 ± 112.91	24.54	0.28
BUN, mmol/L	1.15 ± 0.25 ^b^	1.70 ± 0.42 ^ab^	1.96 ± 0.41 ^a^	1.94 ± 0.7 ^a^	0.12	0.03
GPT, U/L	13.98 ± 1.18	13.95 ± 2.5	13.35 ± 1.67	13.00 ± 0.22	0.32	0.69
GOT, U/L	62.28 ± 8.54	58.90 ± 4.44	58.82 ± 16.69	65.79 ± 16.67	2.82	0.84
MDA, nmol/ml	3.78 ± 0.62 ^c^	7.37 ± 0.44 ^a^	6.18 ± 0.46 ^b^	5.78 ± 0.82 ^b^	0.36	<0.01
T-AOC, U/ml	2.76 ± 0.08 ^a^	1.92 ± 0.21 ^c^	2.01 ± 0.3 ^bc^	2.32 ± 0.16 ^b^	0.10	<0.01
SOD, U/ml	311.51 ± 47.3 ^a^	210.15 ± 21.01 ^b^	284.42 ± 23.37 ^a^	313.21 ± 36.62 ^a^	12.87	<0.01
28d						
TP, ug/ml	51.89 ± 8.39 ^a^	28.68 ± 4.77 ^b^	37.59 ± 14.89 ^ab^	39.94 ± 2.02 ^ab^	3.17	0.02
AIB, g/L	10.30 ± 0.45 ^a^	8.53 ± 0.28 ^b^	9.44 ± 1.33 ^ab^	10.15 ± 1.08 ^a^	0.26	0.04
UA, μmol/ml	195.39 ± 90.95	215.08 ± 66.77	176.02 ± 27.73	180.14 ± 22.33	11.71	0.66
BUN, mmol/L	0.94 ± 0.52	1.18 ± 1.73	0.47 ± 0.34	1.20 ± 1.61	0.26	0.73
GPT, U/L	12.75 ± 0.12	12.94 ± 0.23	12.75 ± 0.18	12.8 ± 0.24	0.04	0.38
GOT, U/L	52.49 ± 1.16	50.33 ± 4.9	52.61 ± 9.21	54.44 ± 2.53	1.19	0.74
MDA, nmol/ml	2.06 ± 0.95 ^c^	5.94 ± 0.67 ^a^	4.60 ± 1.02 ^b^	4.27 ± 1.09 ^b^	0.39	<0.01
T-AOC, U/ml	2.62 ± 0.12 ^a^	1.64 ± 0.08 ^b^	1.79 ± 0.21 ^b^	1.71 ± 0.03 ^b^	0.10	<0.01
SOD, U/ml	295.61 ± 11.33 ^a^	227.86 ± 20.38 ^b^	271.36 ± 30.32 ^a^	280.74 ± 23.46 ^a^	8.28	<0.01

Abbreviations: CON, control group birds fed a basal diet; CPEA, birds fed a basal diet and infected with Clostridium perfringens and Eimeria acervuline; CPEA-EO350, birds fed a basal diet supplemented with 350 mg/kg essential oils and infected with Clostridium perfringens and Eimeria acervuline; CPEA-EO500, birds fed a basal diet supplemented with 500 mg/kg essential oils and infected with Clostridium perfringens and Eimeria acervuline; TP, total protein; ALB, albumin; UA, uric acid; BUN, urea nitrogen; ALT, alanine aminotransferase; AST, aspartate aminotransferase; SOD, superoxide dismutase; T-AOC, total antioxidant capacity; MDA, malondialdehyde. ^a–c^ Means within a column lacking a common superscript differ as per the corresponding *p* value indicated in the *p* value row.

## Data Availability

The data presented in this study are available on request from the corresponding author.

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
