# Peer review of "Complex of Lauric Acid Monoglyceride and Cinnamaldehyde Ameliorated Subclinical Necrotic Enteritis in Yellow-Feathered Broilers by Regulating Gut Morphology, Barrier, Inflammation and Serum Biochemistry"

_animals, 2023, doi:10.3390/ani13030516_

Round 1

Reviewer 1 Report

This article is generally interesting and my comments aim to increase the scientific soundness and clarity of it. English grammar and syntax in the manuscripts must be checked and corrected by a native English-speaking person.

Line 27 and throughout the text – Phyla names should be written in italics.

Line 41 and throughout the text – “Please change to TNF- α”.

Line 87-90 – Please explain your hypothesis properly.

Line 137 – in fact the authors measured only villi height and crypts depths. Therefore, the term “intestinal histomorphology” is not justified.

Line 139 – please provide method of euthanasia.

Figure 2 – scale bars are missing. All images are without any description.

Lines 341-343 – The authors should mention that some enzymes (like xylanse) or combination of enzymes with dietary components (xylanase + hybrid rye) can affect the chicken intestinal barrier (tight junction proteins) (see. Donaldson et al. Modern Hybrid Rye, as an Alternative Energy Source for Broiler Chickens, Improves the Absorption Surface of the Small Intestine Depending on the Intestinal Part and Xylanase Supplementation. Animals 2021, 11, 1349.). This issue is definitely in line with the topic of the current article and in Reviewers opinion the authors have to discuss it briefly and acknowledge this work.

Line 403 – In the present form the conclusions are just descriptive repetition of results. Some future perspectives or recommendations are needed. Please explain what “better” means (line 410). In what sense “better”?

Author Response

Dear Reviewer,

Thank you for your letter and for the reviewers’ comments concerning our manuscript entitled“Complex of Lauric Acid Monoglyceride and Cinnamaldehyde Ameliorated Subclinical Necrotic Enteritis in Yellow-feathered Broilers by Regulating Gut Morphology, Barrier, Inflammation and Serum Biochemistry”(ID: animals-2157014).Those comments are all valuable and very helpful for revising and improving our paper, as well as the important guiding significance to our researches. We have studied comments carefully and have made correction which we hope meet with approval. Revised portion are marked in red in the paper.The main corrections in the paper and the responds to the reviewer’s comments are as flowing:

Responds to the reviewer’s comments:

Comment 1: Line 27 and throughout the text – Phyla names should be written in italics.

Response: Thank you for your professional reminders, we have written the full text - door name correction in italics.

Comment 2:Line 41 and throughout the text – “Please change to TNF- α”

Response: Thank you for your professional reminders, We have changed the full text "TFN-α" to "TNF-α".

Comment 3:Line 87-90 – Please explain your hypothesis properly.

Response: Thanks for your professional advice, After reviewing this sentence carefully, we believe that the description is wrong, and we will rewrite lines 87 - 90 and hope to get your approval. (Line 85-96)

Comment 4:Line 137 – in fact the authors measured only villi height and crypts depths. Therefore, the term “intestinal histomorphology” is not justified.

Response: Thanks for your professional advice,After much thought, we have changed "intestinal histomorphology" to "Intestinal tissue section" and hope to get your approval.

Comment 5:Line 139 – please provide method of euthanasia.

Response: Thanks for your professional advice, We have added the method of euthanasia to line 170-171. We performed euthanasia with an artificial neck dislocation (breaking the connection between the head and the cervical spine to achieve rapid death).

Comment 6:Figure 2 – scale bars are missing. All images are without any description.

Response: Thanks for your professional advice, We have included the scale-up description of Figure 2 in the caption below the image. Other images in the article are supplemented with a caption below the image.

Comment 7:Lines 341-343 – The authors should mention that some enzymes (like xylanse) or combination of enzymes with dietary components (xylanase + hybrid rye) can affect the chicken intestinal barrier (tight junction proteins) (see. Donaldson et al. Modern Hybrid Rye, as an Alternative Energy Source for Broiler Chickens, Improves the Absorption Surface of the Small Intestine Depending on the Intestinal Part and Xylanase Supplementation. Animals 2021, 11, 1349.). This issue is definitely in line with the topic of the current article and in Reviewers opinion the authors have to discuss it briefly and acknowledge this work.

Response: Thank you very much for your valuable advice. We have looked up the article(Donaldson et al. Modern Hybrid Rye, as an Alternative Energy Source for Broiler Chickens, Improves the Absorption Surface of the Small Intestine Depending on the Intestinal Part and Xylanase Supplementation. Animals 2021, 11, 1349.), This article is considered to be in line with the topic of the article and is briefly discussed and inserted in lines 475-476.

Comment 8:Line 403 – In the present form the conclusions are just descriptive repetition of results. Some future perspectives or recommendations are needed. Please explain what “better” means (line 410). In what sense “better”?

Response: Thank you very much for your patiently advice. We have revised the conclusion of the paper and put forward some future views and suggestions for this study. 500mg/kg concentration was better than 350mg/kg concentration in improving performance, intestinal pathological damage score and intestinal morphology convenience of yellow-feathered broilers.

Special thanks to you for your good comments. 

We tried our best to improve the manuscript and made some changes in the manuscript.  These changes will not influence the content and framework of the paper. And here we did not list the changes but marked in red in revised paper.

We appreciate for Editors and Reviewers’ warm work earnestly, and hope that the correction will meet with approval.

Once again, thank you very much for your comments and suggestions.

Reviewer 2 Report

The manuscript is excellent; with interesting results for poultry. I really liked the text, with a logical sequence; however, two parts could be improved.

1) the authors' hypothesis for the tested combination is not clear; what "technically" led the authors to consider this combination of ingredients capable of interfering with intestinal health, inflammatory response, among others. This has to be made clear in the introduction section.

2) the conclusion was very similar to the description of the results section. Attention, for conclusion you must use the results to make an interpretation of what happened and thus conclude something. Rewrite that section.

Author Response

Dear Reviewer,

Thank you for your letter and for the reviewers’ comments concerning our manuscript entitled“Complex of Lauric Acid Monoglyceride and Cinnamaldehyde Ameliorated Subclinical Necrotic Enteritis in Yellow-feathered Broilers by Regulating Gut Morphology, Barrier, Inflammation and Serum Biochemistry”(ID: animals-2157014).Those comments are all valuable and very helpful for revising and improving our paper, as well as the important guiding significance to our researches. We have studied comments carefully and have made correction which we hope meet with approval. Revised portion are marked in red in the paper.The main corrections in the paper and the responds to the reviewer’s comments are as flowing:

Responds to the reviewer’s comments:

Comment 1:the authors' hypothesis for the tested combination is not clear; what "technically" led the authors to consider this combination of ingredients capable of interfering with intestinal health, inflammatory response, among others. This has to be made clear in the introduction section.

Response: Thanks for your professional advice, We have made improvements in the introduction section, because the single application of monoglyceride laurate and cinnamaldehyde in the performance and antibacterial aspects of yellow-feathered broilers has achieved good results, because there are also studies showing that the synergistic effect of the combination of two or more compound components is greater than the sum of effects of each single compound at the same dose. Therefore, this study speculated that monoglyceride laurate and cinnamaldehyde may affect the growth and intestinal health of broilers. Line85-96

Comment 2:the conclusion was very similar to the description of the results section. Attention, for conclusion you must use the results to make an interpretation of what happened and thus conclude something. Rewrite that section.

Response: Thank you very much for your valuable advice, We have rewritten the conclusion in combination with the suggestions of the three reviewers(Line558-565). 

Special thanks to you for your good comments. 

We tried our best to improve the manuscript and made some changes in the manuscript.  These changes will not influence the content and framework of the paper. And here we did not list the changes but marked in red in revised paper.

We appreciate for Editors and Reviewers’ warm work earnestly, and hope that the correction will meet with approval.

Once again, thank you very much for your comments and suggestions.

Reviewer 3 Report

The study presented the results about the effects of dietary supplenmation with lauric acid monoglyceride (organic acids derivatives) and cinnameldehyde (a kind of essential oil) on growth perfroamnce, intestinal barrier function and mucosal immune responses ae well as serum antioxidamnt functions of yellow-feathered broilers inflicted with necrotic enteritis (NE) (co-infected with Eimeria acervulina (EA) and Clostridium perfringens (CP)). This study is very much helpful for non-antibiotics prevention and controll of NE in yellow-feathered broiler chickens. However, some minor errors and question will need to be answered and revised until the paper could be accepted for publication.

1) Does lauric acids belong to essential oil or organic acids? what is the difference between lauric acids and lauric essential oils?  please specify.

2) please italicize both Eimeria acerulina and Clostridium perfringens in this article (including tables and figures).

3) please changed all "challenged or infected with Clostridium perfringens and Eimeria acerulina" into "challenged or infected with Eimeria acerulina and Clostridium perfringens".

4)please checked the presentation of all data in all tables, all data (should be accurated to two decimal  places, for example, in table 4 and table 5, 1 +/- 0  should be changed into 1.00 +/- 0.00, etc. 

5) In table 2,  please add 5‘---3’ after "sequence" , add F: and R: in front of every gene sequence.

6) In Figure 1 and Figure 2, please add a letter (a, b, c, ...... or A, B, C, ......) next to each figure to distinguish them.

7)  In line 80 and 81, two Eschichia coli, please delete one. what is fecal Eschichia coli?  fecal should not be italic.

8) please revised this incorrect sentences (line 87-90).

9) please check feed formula (Table 1). Two "limestone 2.00"?  what is the "M+C" or "CP" ? please expressed M+C, CP  in full name rather than abbreviation.

10) In line 166-167, please write full name of all genes.

11) Is the significant P italic or not? please checked them and expressed it at the same type.

12) please rewrite this paragraph (lines 272-293).

13) please checked all punctuation mark of this paper (such as, .. or .)

14) please standardize the presentation of the quoated literature in the article. For example, Shirani, Jazi [31], Mohebodini, Jazi [32], etc.

15) in lines 364-372, more sentences unrelated with this study were presented in this article, reviewer suggested that these sentences should be revised and shortened.

16)please give a accurate conclusion based on all findings from this study. why is 500 mg/kg better than 350 mg/kg?

Author Response

Dear Reviewer,

Thank you for your letter and for the reviewers’ comments concerning our manuscript entitled“Complex of Lauric Acid Monoglyceride and Cinnamaldehyde Ameliorated Subclinical Necrotic Enteritis in Yellow-feathered Broilers by Regulating Gut Morphology, Barrier, Inflammation and Serum Biochemistry”(ID: animals-2157014).Those comments are all valuable and very helpful for revising and improving our paper, as well as the important guiding significance to our researches. We have studied comments carefully and have made correction which we hope meet with approval. Revised portion are marked in red in the paper.The main corrections in the paper and the responds to the reviewer’s comments are as flowing:

Responds to the reviewer’s comments:

Comment 1:Does lauric acids belong to essential oil or organic acids? what is the difference between lauric acids and lauric essential oils?  please specify.

Response :Thank you for your question, Lauric acid formula C12H24O2, lauric acid monoglyceride formula C15H30O4 are medium chain fatty acid derivatives, belong to organic acids. Lauric acid is an organic acid with the formula C12H24O2, and lauric essential oil is an essential oil extracted from the plant laurel, which contains lauric acid.

Comment 2:please italicize both Eimeria acerulina and Clostridium perfringens in this article (including tables and figures).

Response : Thank you for your professional reminders. In this paper, Eimeria acerulina and Clostridium perfringens have been modified in italics.

Comment 3:please changed all "challenged or infected with Clostridium perfringens and Eimeria acerulina" into "challenged or infected with Eimeria acerulina and Clostridium perfringens".

Response : Thanks for your professional advice, We have changed all "challenged or infected with Clostridium perfringens and Eimeria acerulina" to "challenged or infected with Eimeria acerulina and Clostridium perfringens".

Comment 4:please checked the presentation of all data in all tables, all data (should be accurated to two decimal  places, for example, in table 4 and table 5, 1 +/- 0  should be changed into 1.00 +/- 0.00, etc. 

Response : Thank you for your professional reminders. We have accurate all the data in the paper to two decimal places.

Comment 5:In table 2,  please add 5‘---3’ after "sequence" , add F: and R: in front of every gene sequence.

Response : Thank you for your professional reminders. We have modified Table 2.

Comment 6:In Figure 1 and Figure 2, please add a letter (a, b, c, ...... or A, B, C, ......) next to each figure to distinguish them.

Response : Thank you very much for your professional advice. We have changed Figures 1 and 2.

Comment 7:In line 80 and 81, two Eschichia coli, please delete one. what is fecal Eschichia coli?  fecal should not be italic.

Response : Thank you for your professional reminders , We have removed the excess Eschichia coli.  Fecal coliform is a group of intestinal bacteria growing in the intestines of humans and warm-blooded animals. It is excreted with feces, accounting for more than 1/3 of the dry weight of feces, so it is called fecal coliform.

Comment 8:please revised this incorrect sentences (line 87-90).

Response : Thank you for your professional reminders , We have corrected line 87-90. (Line93-96)

Comment 9:please check feed formula (Table 1). Two "limestone 2.00"?  what is the "M+C" or "CP" ? please expressed M+C, CP  in full name rather than abbreviation.

Response :  Thank you for your professional reminders , we removed the extra "limestone 2.00". "M+C" is Methionine and Cysteine, and "CP" is CrudeProtein. We've represented it in full terms in the table.

Comment 10:In line 166-167, please write full name of all genes.

Response : Thank you very much for your patiently advice. We've written down the full names of all the genes on lines 166 -167. (Line197-199)

Comment 11:Is the significant P italic or not? please checked them and expressed it at the same type.

Response : Thank you for your professional reminders ,We have reviewed the full text and changed the p to italic.

Comment 12:please rewrite this paragraph (lines 272-293).

Response : Thank you very much for your patiently advice. We've already rewritten this paragraph. Line366-379

Comment 13:please checked all punctuation mark of this paper (such as, .. or .)

Response :  Thank you for your professional reminders ,We have corrected the punctuation in the whole article.

Comment 14:please standardize the presentation of the quoated literature in the article. For example, Shirani, Jazi [31], Mohebodini, Jazi [32], etc.

Response : Thank you for your professional reminders ,We have corrected the presentation of the references cited in the article.

Comment 15:in lines 364-372, more sentences unrelated with this study were presented in this article, reviewer suggested that these sentences should be revised and shortened.

Response :Thank you very much for your valuable advice. We have removed some sentences that are not relevant to this study. Line 481

Comment 16:please give a accurate conclusion based on all findings from this study. why is 500 mg/kg better than 350 mg/kg?

Response : Thank you for your professional reminders , We have rewritten the conclusion in combination with the suggestions of the three reviewers(Line558-565). 500mg/kg concentration was better than 350mg/kg concentration in improving performance, intestinal pathological damage score and intestinal morphology convenience of yellow-feathered broilers.

Special thanks to you for your good comments. 

We tried our best to improve the manuscript and made some changes in the manuscript.  These changes will not influence the content and framework of the paper. And here we did not list the changes but marked in red in revised paper.

We appreciate for Editors and Reviewers’ warm work earnestly, and hope that the correction will meet with approval.

Once again, thank you very much for your comments and suggestions.

Round 2

Reviewer 1 Report

The authors reasonably answered to my concerns.

Reviewer 3 Report

The reviewer has no commentes and suggestions for authors and this paper.